# Metabolic Treatment of Wolfram Syndrome

**DOI:** 10.3390/ijerph19052755

**Published:** 2022-02-27

**Authors:** Dario Iafusco, Angela Zanfardino, Alessia Piscopo, Stefano Curto, Alda Troncone, Antonietta Chianese, Assunta Serena Rollato, Veronica Testa, Fernanda Iafusco, Giovanna Maione, Alessandro Pennarella, Lucia Boccabella, Gulsum Ozen, Pier Luigi Palma, Cristina Mazzaccara, Nadia Tinto, Emanuele Miraglia del Giudice

**Affiliations:** 1Regional Center of Pediatric Diabetology “G.Stoppoloni”, University of Campania “Luigi Vanvitelli”, 80138 Naples, Italy; angela.zanfardino@unicampania.it (A.Z.); alessia.piscopo@hotmail.it (A.P.); stecurto@yahoo.it (S.C.); alda.troncone@unicampania.it (A.T.); antonietta.chianese1@gmail.com (A.C.); serenarollato@gmail.com (A.S.R.); veronicatesta92@gmail.com (V.T.); alessandro.pennarella@studenti.unicampania.it (A.P.); lucia.boccabella@hotmail.it (L.B.); ozen_gulsum@hotmail.com (G.O.); pieropalma2710@gmail.com (P.L.P.); emanuele.miragliadelgiudice@unicampania.it (E.M.d.G.); 2Department of Molecular Medicine and Medical Biotechnology, University of Naples “Federico II”, 80131 Naples, Italy; iafusco@ceinge.unina.it (F.I.); maione@ceinge.unina.it (G.M.); cristina.mazzaccara@unina.it (C.M.); nadia.tinto@unina.it (N.T.); 3CEINGE Advanced Biotechnologies, 80131 Naples, Italy

**Keywords:** Wolfram Syndrome, diabetes mellitus, insulin therapy

## Abstract

Wolfram Syndrome (WS) is a very rare genetic disorder characterized by several symptoms that occur from childhood to adulthood. Usually, the first clinical sign is non-autoimmune diabetes even if other clinical features (optic subatrophy, neurosensorial deafness, diabetes insipidus) may be present in an early state and may be diagnosed after diabetes’ onset. Prognosis is poor, and the death occurs at the median age of 39 years as a consequence of progressive respiratory impairment, secondary to brain atrophy and neurological failure. The aim of this paper is the description of the metabolic treatment of the WS. We reported the experience of long treatment in patients with this syndrome diagnosed in pediatric age and followed also in adult age. It is known that there is a correlation between metabolic control of diabetes, the onset of other associated symptoms, and the progression of the neurodegenerative alterations. Therefore, a multidisciplinary approach is necessary in order to prevent, treat and carefully monitor all the comorbidities that may occur. An extensive understanding of WS from pathophysiology to novel possible therapy is fundamental and further studies are needed to better manage this devastating disease and to guarantee to patients a better quality of life and a longer life expectancy.

## 1. Introduction

Wolfram Syndrome (WS) is a very rare genetic disorder, autosomal recessive, and characterized by many symptoms (diabetes mellitus, diabetes insipidus, optic atrophy, neurosensorial deafness and neurological signs) that appear in succession from childhood to adulthood [1]. Other clinical features include bladder and bowel dysfunction, peripheral neuropathy, psychiatric and neurological abnormalities.

### 1.1. Pathogenesis

Two disease genes have been identified: *WFS1* (4p16.1) and *CISD2* (4q24). Mutations in the *WFS1* gene cause more than 90 percent of Wolfram Syndrome type 1 (WS1) cases. *WFS1* encodes wolframine, a protein located in the endoplasmic reticulum (ER), which plays a role in calcium homeostasis and in the response to unfolded proteins. It is highly expressed in the brain tissue, pancreatic β-cells, heart, lung, and placenta. *CISD2*, responsible for Wolfram Syndrome type 2 (WS2), encodes the endoplasmic reticulum intermembrane small protein (ERIS), a small intermembrane protein which plays an important role in ER, mitochondria membrane integrity, and in the functional cross-talk between these two cellular compartments [1].

The large parts of mutations in these genes has an autosomal recessive mode of transmission, however, autosomal dominant mutations have been described in association with WS-like disease [1].

It has been established that ER dysfunction and ER stress are critical pathogenic components of WS, and in fact, *WFS1* mutations can provoke elevated ER stress levels which lead to β-cells dysfunction and cell death [2].

### 1.2. Metabolic Features of Wolfram Syndrome

Wolfram Syndrome (WS) is a very rare genetic disorder characterized by several symptoms, such as juvenile-onset diabetes mellitus, diabetes insipidus, optic nerve atrophy, hearing loss, and neurodegeneration, which appear from childhood to adulthood [1].

The prognosis of this syndrome is currently poor, as most patients die prematurely with severe neurological disabilities without effective treatments that can delay or reverse the progression of the disease. The use of careful clinical monitoring and supportive care can help relieve the suffering of patients and improve their quality of life [2].

Non-autoimmune and Insulinopenic Diabetes Mellitus is generally the first symptom diagnosed around the age of six (range: 3 weeks to 16) years old, while other alterations manifest later in life. Optic atrophy (median age of 11 years), sensorineural deafness (median age of 20 years) and neurological abnormalities (median age of 30 years) occur in a significant fraction of affected individuals [3].

In our casuistic of 15 patients diagnosed in the Regional Centre of Pediatric Diabetology from 1986 to 2015, the age at the onset of diabetes ranged between 2 to 11 years old with a median of 5 years, as shown in Table 1.

Very intriguing is the patient 14 who had congenital deafness treated with cochlear implantation. Only the onset of associated non-autoimmune diabetes revealed the diagnosis of WS.

Compared to type 1 autoimmune diabetes (T1D), ketoacidosis at the onset of diabetes is rare in Wolfram Syndrome even if it is not excluded and the patients have a longer duration of the remission period, lower insulin requirements and a good metabolic control as demonstrated by the levels of HbA1c. In our cases the mean of HbA1c was 7.5% (range 6–9.3%) and the mean dose of insulin during the entire follow up ranged from 0.3 Units/Kg/day to 0.8 Units/Kg/day as well as the dose of Type 1 diabetic patients during the period of partial remission (“honey moon”). A feature that could differentiate the insulin therapy in WS compared to T1D autoimmune patients, in which there is an acute destruction of β cells, could also be the lack or, on the contrary, the extension of the honey moon period. Some WS patients have been descripted as showing a remarkably long remission period of 8 years or an insulin requirement of 0.5 IU/kg/day [1]. In addition, during the pubertal period, the increase of the insulin dosage may not be so high as in T1D. This feature could also be due to the lack of severe insulin resistance, the characteristic of puberty, and due to the reduction of sex hormones, which is found in this syndrome.

A correlation between metabolic control of diabetes, the number of associated symptoms and the progression of the neurodegenerative component of the disease has been described [1].

There is a correlation between postprandial hyperglycemic spikes and the progression of the neurodegenerative component of the disease. This would also be explainable by the causative ER stress pathogenesis theory. There is also a study that identifies glucose toxicity as an accelerating feature in the progression of the disease [4].

On the other hand, also in Type 1 diabetes, a history of chronic hyperglycemia appears to be more injurious than previously suspected [5]. Neurocognitive deficits manifest across multiple cognitive domains, including executive function and speed of information processing. One study showed that glucose variability might have a greater adverse impact on the developing brain than either prolonged high or low glucose levels [6]. There were two possible mechanisms through which blood glucose variability leads to cognitive dysfunction, according to previous studies. One showed that glycemic variability was associated with increased production of reactive oxygen species that could damage the central nervous system [7]. Another mechanism showed that oscillating glucose could have a more toxic impact on oxidative stress generation than constant high glucose, which might lead to mitochondrial dysfunction and neuronal cell damage [8].

Compared to other forms of non-autoimmune pediatric diabetes in which the comparison between glycaemia and HbA1c could simplify the diagnosis [9], in WS there are no similar data and/or experiences, and only the negativity of autoimmune diabetes antibodies together with the appearance of other clinical symptoms can induce to carry out genetic analysis. The differential diagnosis has to be done with other syndromes as, for example, Alstrom syndrome [10] or with mithocondrial diseases [11].

In this regard, in the last decade, the Next Generation Sequencing (NGS) has proved to be an effective molecular diagnostic strategy showing good diagnostic sensitivity, especially following a careful clinical selection of patients [12,13,14]. The genetic heterogeneity and complexity that distinguishes the molecular basis of several forms of diabetes, in particular in pediatric age, imposes the need to refer to experts in the field of this disease, who can take advantage of the most advanced sequencing methods, such as NGS, to sequence multiple genes simultaneously, in the effort to reach the maximum genetic yield [15].

The criteria that should prompt for genetic testing are different in pediatrics in respect to adults. In our experience, all the cases of diabetes mellitus developed during childhood or adolescence without diabetes antibodies (GAD and/or IA2 and/or IAA and/or ZnT8) needed to receive the pathogenetic diagnosis by NGS panel. In our center, we have an NGS panel with 41 genes of non- autoimmune diabetes including the Wolfram genes. This method of proceeding with the NGS as the first step after antibodies, is only one modern approach because, before the NGS panels, only the clinic and the association of non-autoimmune diabetes with other symptoms such as, in particular, optic atrophy, could lead to suspect the syndrome. In our experience optic atrophy was an occasional finding during an eye examination done to recognize the early complications of diabetes.

Hearing loss can also be an associated pathology of suspicion and it is certainly very interesting that the peculiar case that we have described in this paper, of a child who was born with congenital sensorineural deafness for which it was necessary to perform the cochlear implant, and who developed diabetes after only a few years, which allowed him to make the diagnosis of genetic syndrome, then confirmed by a genetic analysis.

In WS, due to neurological dysfunctions caused by a perturbation of ER function, episodes of severe hypoglycemia may be more frequent. For this reason, the use of real time Control Glucose Monitoring Systems (CGMS) with the possibility of using alarms, even predictive ones, of hypoglycemia must be strongly recommended in such patients. The alarm must be both audible and vibrating to allow patients to warn them in time.

The relative ease with which good metabolic control is achieved in this type of patient makes microvascular complications much less frequent, and the slower progression of the same could be related to a persistence of residual insulin secretion and total insulin deficiency, which is not as quick as in T1D. The reduced dose of insulin used also makes macrovascular complications highly unlikely. Degeneration of pancreatic β-cells, where Wolframin ER Transmembrane Glycoprotein (WFS1) is highly expressed, determines insulinopenia. However, in literature, cases of co-occurrence of WS and T1D have been reported [16,17,18].

Despite residual c-peptide production, insulin therapy in pediatrics should be considered as the first choice of treatment. However, in deciding which insulin therapy scheme to use, a whole series of variables that are typical of the syndrome must be taken into account.

Gastrointestinal disorders as bowel dysmotility, gastroparesis, and bowel incontinence have been reported [19]. It is very important to be prepared to consider the onset of various complications to adapt insulin therapy to the need. Since neurological anomalies have been described that affect the intestine and affect the absorption times, for example, such as gastroparesis, intestinal transit alterations, etc., it is necessary to adapt the pharmacokinetics of insulin to the absorption times of carbohydrates. The preferred therapy scheme is always the basal bolus one, however, in our experience, in many cases we preferred to use regular human insulin as a bolus rather than a rapid analogue. Therefore, we used the absorption and peak delay to counteract the slowed absorption of carbohydrates due to gastroparesis. Additionally, the high risk of hypoglycemia in patients with neurological problems must be taken in account in the choice of the insulin therapy scheme.

Insulin treatment has to be adapted and modulated to the frequent steroid supplementation that may be necessary for some patients during their severe pulmonary infections [20].

In children and adolescents, insulin therapy represents the first choice of treatment only because many drugs are considered off label in paediatrics (Table 2). Therapy with a subcutaneous insulin pump may also be taken into account in these patients using the usual sites of infusion [21]. In these last years, in fact, many other hypoglycemic drugs may be considered in the treatment of adults with this syndrome [22].

In adults, new therapeutic drugs already used in the treatment of Type 2 Diabetes (T2D) have been under experiment to obtain the best metabolic control in WS as, for example, Glucagon-Like Peptide-1 Receptor (GLP-1R) agonists (a peptide secreted from intestinal L cells after meals) [1,23,24,25] and exenatide (an incretin mimetic agent). The use of these agents has been demonstrated to reduce 70% in the daily insulin doses often associated with better glycemic control in patients with diabetes by WS [1].

Liraglutide also (a long acting agonist of GLP-1R) improves patients’ glycemic control and reduces the daily insulin dose [1]. However, GLP1 reduces pancreatic β-cell apoptosis mediated by ER stress, improves β-cell growth and survival, and also has neuroprotective functions [1,26]. Clinical trials for T2D evidenced that, compared to placebo, these drugs have no effect or even cause a worsening on the progression of diabetic complications therefore their use remains controversial [1]. Inhibitors of dipeptidyl peptidase-4 (DPP-4), an enzyme that deactivates GLP-1, were suggested as possible therapy as they would lead to an increase of GLP-1 [27].

Very intriguing are the descriptions of the effects of liraglutide in preventing diabetes in animal models. In the rats with WS, the treatment with this drug delayed the onset of diabetes and also protected against vision loss [28], and may have a neuroprotective effect [29].

Therefore, early diagnosis and prophylactic treatment with liraglutide may also prove to be a promising treatment option for WS patients by increasing their quality of life.

Although it is very evident that ER dysfunction is involved in WS, the pathogenic mechanisms are not fully understood. Consequently, there are no consolidated therapies available that effectively act on ER dysfunction [30].

Animal models have been experimented with fair success and the Muscarinic receptors 3 agonists and the activation of the muscarinic pathway has been demonstrated to potentiate insulin secretion in mice with WS [31].

In a disease characterized by the progressive degeneration of neurological tissues and β-cells which could be considered nervous tissue, the main goals of the treatment should be to represent the slowdown of progression of the neurodegeneration, and eventually, by the replacement of damaged tissues such as pancreatic β-cells and retinal cells [1]. Very promising is the recent use of Dantrolene sodium, a hydantoin derivative skeletal muscle relaxant, indicated in the treatment of malignant hyperthermia, whose mechanism of action revolves around the inhibition of ryanodine receptors (RyRs) on the ER [32], reducing cytosolic calcium and preserving ER calcium. Interestingly, recent studies have proposed a potential role for dantrolene sodium as a treatment for neurodegenerative disorders, such as Huntington’s disease [33], spinocerebellar ataxia [34], and Alzheimer’s disease [35], where ER calcium may play a role in disease pathogenesis. Important results have been demonstrated in mice and induced pluripotent stem cell (iPSC) models of WS treated with dantrolene sodium [36]. An important clinical trial has assessed the effects of Dantrolene sodium on residual pancreatic β-cell function, visual acuity, neurological function, and quality-of-life measures in pediatric and adult subjects with WS [37].

Mice model research could reveal a suppression of the apoptosis rate and restoration of dysfunctional β-cells for WFS by using Dantrolene sodium [38].

Valproate-Natrium also seems to attenuate endoplasmic reticulum stress-induced apoptosis in WS [39] and there are in vitro [40] studies that confirmed its effect.

### 1.3. Wolfram Syndrome Treatment by Regenerative and Gene Therapy

In the future, the WS treatment will be represented by regenerative and gene therapy which may lead to the replacement of damaged tissues (pancreatic β-cells and retinal cells) [1]. To this purpose, several studies are ongoing in Urano’s laboratory, at the Washington University School of Medicine, which is one of the most important worldwide centers on WS [41]. In particular, the main aim should be to obtain induced pluripotent stem cells (iPS) from skin cells of patients with WS, to be differentiated into neurons, retinal cells and β-cells, and thus to be used for transplantation [20,42]. Gene therapies based on adeno-associated virus and Clustered Regularly Interspaced Short Palindromic Repeats (CRISPR) technology are intensively being studied to correct *WFS1* mutations [43]. Gene therapy is also investigated to induce the production of mesencephalic astrocyte-derived neurotrophic factors (MANF) in WS patients. In fact, MANF is a regeneration factor produced by astrocytes that can prevent cell death and activate the proliferation of remaining β-cells, neurons, and retinal ganglion cells by leveraging the natural ability of the human body to regenerate damaged tissues [44].

A recent study reported gene therapy to cure WS. Maxwell et al. [45,46] corrected a *WFS1* pathogenic variant in patient fibroblast-derived induced pluripotent stem cells (iPSCs) by using CRISPRCas9, which were differentiated into β-cells and transplanted into β-cell depleted mice, leading to a restart of insulin production and blood glucose regulation. The gene-corrected β-cells showed improved glucose-stimulated insulin secretion and reversed hyperglycemia for 6 months after their transplantation into diabetic mice. This study may open up the possibility of autologous β-cell transplants for patients with WS. All these findings demonstrate the possibility to treat WS by gene therapy, and encourage further research for a specific treatment of patients, i.e., by mainly targeting and solving the ER stress [47]. However, further studies are needed to confirm the validity of these findings.

It is important to underline that for all rare diseases, as for example WS, international consortia and networks are a source of important research and therapeutic trials.

For WS we have the SNOw Foundation and the Eyes Hope Foundation working for better understanding and for new therapeutic options in patients with this disease. Furthermore, ENDO-ERN and EURRECA are two EU-funded networks that aim to promote knowledge sharing, education and research on rare endocrine diseases [22].

## 2. Conclusions

Wolfram Syndrome is a very rare genetic disorder characterised by many symptoms that appear one after the other in succession, from childhood to adulthood. Usually, non-autoimmune Diabetes Mellitus is recognized early also if other features (optic subatrophy, deafness, diabetes insipidus) may be present in an early state and may be diagnosed after Diabetes Mellitus onset. Neurological involvement appears later.

Insulin is still the first line therapy at the onset of diabetes even if new drugs are appearing on the scientific landscape of this complex syndrome. The associated pathologies influence the therapeutical choices. Prognosis remains poor, the death occurs at the median age of 39 years but it is the consequence of progressive respiratory impairment secondary to brain atrophy and neurological failure. Many hopes have to be entrusted to gene therapy.

## Figures and Tables

**Table 1 ijerph-19-02755-t001:** Casuistic of Wolfram Syndrome patients of the Regional Center of Paediatric Diabetology “G. Stoppoloni” of University of Campania “Luigi Vanvitelli”.

**Patients**	**1**	**2**	**3**	**4**	**5**	**6**	**7**	**8**	**9**	**10**	**11**	**12**	**13**	**14**	**15**
Sex	F	M	F	F	M	M	M	M	M	M	M	F	M	F	F
Age actual (years)	27.3	28.3	Dead 41	Lost °	Lost °	Lost °	Lost °	Lost °	Lost °	Lost °	20.1	30.7	29	17.7	21
**Diabetes Mellitus**	YES	YES	YES	YES	YES	YES	YES	YES	YES	YES	YES	YES	YES	YES	YES
Age at onset (years)	5	9	8	3	5	5	4	5	4	3	5	7	7	11	2
HbA1c at onset (%)	12.3	9.9	13					10	11.3	10.6	12	10,6			
Follow up HbA1c mean	8.5	7	9.3	8	6.5	9	8.5	7.5	6.1	6	7.5	6	8.5	6.4	8
Therapy	Insulin	Insulin	Insulin	Insulin	Insulin	Insulin	Insulin	Insulin	Insulin	Insulin	Insulin	Insulin	Insulin	Insulin	Insulin
Daily Insulin Dose (U/Kg/day)	0.5	0.4	0.8	0.4	0.4	0.8	0.7	0.3	0.3	0.4	0.4	0.3	0.7	0.3	0.5
**Diabetes Insipidus**	YES	YES	NO	YES	NO	YES	YES	YES	NO	NO	YES	YES	YES	NO	NO
Age at onset (years)	5	10		23			8	10			7	11	13		
**Eye Disease**	YES	YES	YES	YES	YES	YES	YES	YES	YES	YES	YES	YES	YES	NO	YES
Age at onset (years)	5	9	10	8	10	11	13	16	9	6	18	11	10		12
Optic nerve subatrophy		YES	YES										YES		
Optic nerve atrophy	YES			YES	YES	YES	YES	YES	YES	YES	YES	YES			YES
**Ear Disease**	YES	YES	YES	YES	YES	YES	YES	YES	NO	NO	YES	YES	YES	YES	NO
Age at onset (years)	6	9	15	8	20	20	6	13			4	12	10	At birth Coclear Implantation	
**Urinary Disease**	NO	NO	YES	NO	YES	YES	YES	NO	NO	NO	YES	YES	YES	YES	NO
Press on bladder to pee			YES											YES	
Auto-catheterism					YES	YES	YES				YES	YES	YES		
**Neurological Disease**	YES	YES	YES	YES	YES	YES	YES	NO	NO	NO	NO	YES	YES	NO	NO
Nistagmus	NO	YES	YES	YES	YES	YES	YES					YES	YES		
Vestibular syndrome	YES	NO	YES	YES	YES	YES	YES					YES	YES		
**Patients**	**1**	**2**	**3**	**4**	**5**	**6**	**7**	**8**	**9**	**10**	**11**	**12**	**13**	**14**	**15**
Hypotonia	YES	NO	YES	YES	YES	YES	YES					YES	YES		
Hyporeflexia	YES	NO	YES	YES	YES	YES	YES					YES	YES		
Intellectual impairement	YES	NO	NO	NO	NO	YES	NO					NO	NO		
**Gastrointestinal**	NO	NO	NO	NO	NO	NO	YES	NO	YES	NO	NO	NO	NO	NO	NO
Chronic diarrhea							NO		YES						
Stipsis							YES		NO						
Peptic ulcer							YES		NO						
**Familiar Anamnesis**															
Early death/abortion	YES	YES		YES		YES	YES					YES	YES		
**Genetic Informations**															
WSF1 mutation	p.Gly695 ValHomozygote	p.Tyr528Term Heterozygote									p.Tyr699Cys Homozygote	p.Gly107Arg Homozygote	p.Gly107Arg Homozygote	p.Ala684Val/ IVS6 + 16G > A Compound Heterozygote	p.Tyr291 TermHomozygote

° Lost during the follow up after the transition from our Pediatric Diabetology Center to the Center(s) of care of diabetes of adult.

**Table 2 ijerph-19-02755-t002:** New therapeutic strategies in Wolfram Syndrome.

Therapeutical Options	Effects in WS
Liraglutide (Glucagon Like Peptide 1 Receptor Agonist)	Improve patients’ glycemic control and reduces the daily insulin dose Reduce pancreatic β Cell apoptosis Improve β cell grows and survival Delay the onset of diabetes Protect against vision lost Neuroprotective effect
Exenatide, (Incretin Mimetic Agent)	Improve patients’ glycemic control and reduces the daily insulin dose
Inibitors of Dipeptidil Peptidase 4 (DPP4)	Increase GLP-1 concentration
Muscarinic receptors-3 agonists	Potentiate insulin secretion in mice
Dantrolene sodium (Inhibition of ryanodine receptors (RyRs) on the ER)	Reduce pancreatic β cell apoptosis and restore dysfunctional β cells in mouse models Induce Pluripotent Stem Cell in mouse model Preserve β Cell residual function, visual acuity and neurological functions in humans: pediatrics and adults
Valproate-Natrium	Attenuate endoplasmic reticulum stress-induced apoptosis in vitro

## Data Availability

The authors confirm that all the data reported in this article can be made available upon reasonable request.

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
