# Peer review of "Metabolic Treatment of Wolfram Syndrome"

_ijerph, 2022, doi:10.3390/ijerph19052755_

Round 1

Reviewer 1 Report

Iafusco et al reported an updated review of the new treatment options for wolfram syndrome. The topic is interesting, otherwise limites studies are currently available. Describing personal case series is not the main focus of the paper and could be deleted

Minor comments

Several spelling errors should be corrected throughout the manuscript

Abbrevistions must be precede by the first capital letter

References should be revised according to the guidelines 

Englisdh should be sunstantially revised

Author Response

Dear Reviewer, 

we want to thank for the very useful comments.

1) Iafusco et al reported an updated review of the new treatment options for wolfram syndrome. The topic is interesting, otherwise limites studies are currently available. Describing personal case series is not the main focus of the paper and could be deleted

Response: Also if we agree with you that our paper has to be considered a "review" on this topic, we have described in table 1 our casistic to add a personal experience on the treatment and the long follow up of our many patients. As specify from Reviewer 2 the Wolfram Syndrome is a very rare disorder therefore it may be useful to report our experience considering the large number of cases. 

2) Several spelling errors should be corrected throughout the manuscript

Response: all the spelling errors have been corrected. We have asked to the Journal the editing of our paper.  

3) Abbreviations must be precede by the first capital letter

Response: All the abbrevations have been corrected 

4) References should be revised according to the guidelines 

Response: All the references have been revised according to the guidelines 

5) English should be substantially revised

Response: English has been revised by an USA native speaker and we have asked to the Journal the editing of our paper.

Reviewer 2 Report

This article focuses on metabolic control in patients with Wolfram syndrome (WS). The topic is interesting and important, as WS is a rare disorder and it is therefore helpful to compile the existing knowledge in a review. 

Overall, however, the article lacks some structure in this regard. 
This would benefit from first concisely outlining the metabolic, neurologic, and other symptoms and features that exist in patients with WS. This should be followed by an overview of pathogenesis. This can be followed by a summary of metabolic control - here one's own study can also be included (Table 1) and finally, completed and current pharmacological and gene therapy clinical trials can be reported. 

In the part reporting on metabolic control, concrete figures are missing (how much insulin is little insulin?). Furthermore, differentiation should be made between duration of diagnosis and insulin therapy and age of patients (at least adult/childhood). 

The own study (Table 1) is mentioned. It would be interesting to know the time period over which the data were collected, the duration of the diagnosis and especially the daily insulin dose. Maybe the author has even newer data of a CGM device (time in range, time above goal, time below goal). There are publications that see a correlation between postprandial hyperglycemic spikes and the progression of the neurodegenerative component of the disease. This would also be explainable by causative ER stress pathogenesis theory. A brief discussion on this topic would do the review good. 

The pharmacoloigical studies mentioned are missing the trials with valproate-NAtrium (Tim Barrett laboratory) and with dantrolene sodium. The reference to semaglutide is misplaced, the blindness in WS patients resulting not from retinopathy but from a neurodegenerative disorder of the optic nerve. The landmark results from Finland (Toots M et al 2018, Seppa K et al 2021Jagomae T et al 2021, Seppa K 2019) are not mentioned but are markedly relevant. The off label therapy of Frontino G et al with liraglutide is also important to mention and evaluate. 
Overall, the many foundations working for a better understanding and a new therapeutic option in patients with WS (SNOw Foundation, Eyes Hope Foundation, etc.) could still be mentioned. Furthermore, especially for rare diseases . the existing research networks should be used (ENDO -ERN - Reschke et al. 2021).

The results of gene therapy so far are not as promising as described in the article. In my view, this section is a bit too detailed in terms of the evidence that is still lacking - especially in comparison to the studies that have already taken place and been published. However, it is quite possible that these therapies will play an important role in the future. An outlook is therefore appropriate, but in view of the review character of the article, a retrospective view rather than an overly prospective outlook should be formulated. 

The English varies in parts between British and American, here can still be reworked. Some quotes do not fit the text passages (e.g. quote 22). Some quotes are already somewhat older. Here it should be paid attention to the fact that current quotations are used. 

The descriptions in Figure 1 still need to be optimized. Neither from the text, nor from the illustration it is clear what it is about. 

An overview of metabolic control in WS would also be desirable. Possibly also with a view to the forms of therapy used (CSII, basal bolus, individual insulin doses) and an outlook on which form of therapy has been used best so far. Patients are often dystrophic. Do hypoglycemias play a critical role in therapy ? 

A table or figure about the previous and current trials would also be desirable.

overall I am very happy about the interest in WS - this artiekl can give an important update about the current state of metabolic control in patients with WS after appropriate modulation.

Author Response

Dear Reviewer 2, thanks for your very useful suggestions. 

1) the article lacks some structure in this regard. 
This would benefit from first concisely outlining the metabolic, neurologic, and other symptoms and features that exist in patients with WS. This should be followed by an overview of pathogenesis. This can be followed by a summary of metabolic control - here one's own study can also be included (Table 1) and finally, completed and current pharmacological and gene therapy clinical trials can be reported. 

Response: We thank the Reviewer for these very impressive suggestions. We have rewritten and restructured our paper on the basis of your proposals. We have added a short overview on the pathogenesis that will be detailed furthermore by others Authors in other chapters because our paper is only a part of a monographic number of the Journal entirely dedicated to Wolfram Syndrome.    

2) In the part reporting on metabolic control, concrete figures are missing (how much insulin is little insulin?). Furthermore, differentiation should be made between duration of diagnosis and insulin therapy and age of patients (at least adult/childhood). 

Response: Thank you for your advise, we have added data on insulin therapy in WS patients compared with the insulin dose used in patients with autoimmune disease. Your idea was very intriguing because by the trend of insulin therapy is possible differentiate WS in which there is a slow progressive destruction of beta cells from autoimmune acute lack of secretion.   

3) The own study (Table 1) is mentioned. It would be interesting to know the time period over which the data were collected, the duration of the diagnosis and especially the daily insulin dose.

Response: The time period over which our data were collected is from 1986 to 2015 and we have specified it also in the text. Concerning the duration of the diagnosis, we want to thank for your comment which allows us to clarify that our patients have been followed in pediatrics even over the age of 18 years until the moment of transition to the center of adults. Those "lost to follow up" are patients passed to different centers from our university from which we have not received anymore informations. In the legend of Table 1 we have clarify, following your comment, that our patients have passed to the center for adults not only at 18 years, but also beyond this age.

You know that it is very difficult to obtain from the medical records the insulin dose that changes every day so we have added in the table the mean of the values of daily insulin dose (U/Kg/day) collected duing the visits every three months at the same time of the HbA1c.  

4) Maybe the author has even newer data of a CGM device (time in range, time above goal, time below goal). 

Response: We are sorry, but we have not data of CGM collected until now. If you think that it may be necessary to collect data for publication we can organize it.  

5) There are publications that see a correlation between postprandial hyperglycemic spikes and the progression of the neurodegenerative component of the disease. This would also be explainable by causative ER stress pathogenesis theory. A brief discussion on this topic would do the review good. 

Response: Thank you for this very useful observation. We have added a brief discussion on the glucotoxic effect on the brain function and the possibility that high glycemic variability, in particular hyperglycaemia, appears to be more injurious than previously suspected.   

 6) The pharmacological studies mentioned are missing the trials with valproate-NAtrium (Tim Barrett laboratory) and with dantrolene sodium. The reference to semaglutide is misplaced, the blindness in WS patients resulting not from retinopathy but from a neurodegenerative disorder of the optic nerve. The landmark results from Finland (Toots M et al 2018, Seppa K et al 2021Jagomae T et al 2021, Seppa K 2019) are not mentioned but are markedly relevant. The off label therapy of Frontino G et al with liraglutide is also important to mention and evaluate. 

We are very greatful to Reviewer 2 because thanks to his suggestions we have almost entirely rewritten the paragraph on the Therapy and we have added a new table (table 2).  

7) Overall, the many foundations working for a better understanding and a new therapeutic option in patients with WS (SNOw Foundation, Eyes Hope Foundation, etc.) could still be mentioned. Furthermore, especially for rare diseases . the existing research networks should be used (ENDO -ERN - Reschke et al. 2021).  

Response: We agree with the Reviewer 2 about the importance of the many foundations working for a better understanding and a new therapeutic option in patients with Wolfram Syndrome. At the end of the paragraph on the new therapy trials we have mentioned them.        

The results of gene therapy so far are not as promising as described in the article. In my view, this section is a bit too detailed in terms of the evidence that is still lacking - especially in comparison to the studies that have already taken place and been published. However, it is quite possible that these therapies will play an important role in the future. An outlook is therefore appropriate, but in view of the review character of the article, a retrospective view rather than an overly prospective outlook should be formulated. 

Response: We have a bit modified the paragraph dedicated to the new therapies.

The English varies in parts between British and American, here can still be reworked. Some quotes do not fit the text passages (e.g. quote 22). Some quotes are already somewhat older. Here it should be paid attention to the fact that current quotations are used. 

Response: Thank you for this observation. We have preferred to edit the paper with the specifical service of the Journal. The quotes have been revised and corrected.  

The descriptions in Figure 1 still need to be optimized. Neither from the text, nor from the illustration it is clear what it is about. 

Response: we have deleted the figure 1

An overview of metabolic control in WS would also be desirable. Possibly also with a view to the forms of therapy used (CSII, basal bolus, individual insulin doses) and an outlook on which form of therapy has been used best so far. Patients are often dystrophic. Do hypoglycemias play a critical role in therapy ?

Response: we thank the Reviewer 2 for these important suggestions. We have added news about the therapy of our patients. We have stressed in the test your concept that hypoglycaemia plays a crucial role in the choice of the therapy.    

overall I am very happy about the interest in WS - this artiekl can give an important update about the current state of metabolic control in patients with WS after appropriate modulation.

Response: Thanks for all your suggestions that have ameliorate our paper. 

Reviewer 3 Report

This short paper can be of benefit for the research community in general, but also for practicing endocrinologists facing WS patients in particular. However, the manuscript must be slightly improved/rewritten to meet this aim.

Diagnosis of WS requires genetic testing. I do agree with the authors that next-generation sequencing (NGS) is probably the best technology for this, as there are many mutations in WFS1 (or also CISD2) gene that cause WS. Therefore, authors should elaborate on what is the criteria that should prompt the doctors for genetic testing of WS? Is it lack of antibodies alone or in combination with other symptoms? WS is a nasty disorder and is most certainly underdiagnosed as most of its symptoms are confused with complications of diabetes. Therefore, reccomendations on how to improve diagnostics are needed and should be considered in this paper, given rather large clinical experience of the authors (14 patients with WS). Should all cases of juvenile diabetes go for genetic testing to exclude WS? The earlier it is diagnosed the better for patients. Usually diagnosis is done when optic atrophy is detected in addition to DM, but this is already too late for effective management vision, the largest problem for WS patient. And there are cases of WS without DM.

For the same reason, as diagnosis of WS can be confirmed solely on genetic basis, please provide genetic information of your 14 patients in the table.

The data in the table right now is largely in Yes/No format, it would however be more interesting if it includes more quantitative characteristics as well. 

In addition, if possible update the data with other features often found in WS patients. Is dysarthria/dysphagia present in your patients? Are they heat intolerant? Do they have problems with the heart? Are there any blood abnormalities beside insulin /HbA1c? 

Please provide a comparison of insulin treatment regime between WS patients and autoimmune T1D patients of same age. How much insulin dose is different? Should this lower insulin dose prompt doctors for genetic testing? 

Hypoglycaemia is often a problem for insulin recieving WS patients, please include information on how to prevent this. This is especially true for patients recieving GLP1 RA. 

There are more than one drug currently in clinical trials against WS- valproic acid in UK (prof. T. Barett) and dandrolene in US (prof. Urano), perhaps some other drugs as well. Please provide a short note on these drugs as well, are there any good or should they be considered as failed attemts?

The abstract is too vague, general and repetitive, i would rather see it rewritten to be more concise. There are some spelling and punctuation mistake that need attention. In general, I reccomend this article for publication after the ammendments. 

Author Response

We are very grateful to Reviewer 3 for his considerations that were very useful for us to ameliorate the paper. 

Diagnosis of WS requires genetic testing. I do agree with the authors that next-generation sequencing (NGS) is probably the best technology for this, as there are many mutations in WFS1 (or also CISD2) gene that cause WS. Therefore, authors should elaborate on what is the criteria that should prompt the doctors for genetic testing of WS? Is it lack of antibodies alone or in combination with other symptoms?

Response: This are very arguing considerations of Reviewer 2 and we have added them to the paper.

The criteria that should prompt the Doctors for genetic testing are different in pediatrics respect to adults world. In our experience all the cases of diabetes mellitus developed during childhood or adolescence without diabetes antibodies (GAD and/or IA2 and/or IAA and/or ZnT8) need to receive the pathogenetic diagnosis by NGS panel. In our Center we have an NGS panel with 41 genes of non autoimmune diabetes including the Wolfram genes.    

WS is a nasty disorder and is most certainly underdiagnosed as most of its symptoms are confused with complications of diabetes. Therefore, reccomendations on how to improve diagnostics are needed and should be considered in this paper, given rather large clinical experience of the authors (14 patients with WS). Should all cases of juvenile diabetes go for genetic testing to exclude WS? The earlier it is diagnosed the better for patients. Usually diagnosis is done when optic atrophy is detected in addition to DM, but this is already too late for effective management vision, the largest problem for WS patient. And there are cases of WS without DM.

Response: Of course, this way to proceeding with the NGS as first step after antibodies, is only the modern one approach because, before the NGS panels, only the clinic and the association of non-autoimmune diabetes with other symptoms such as, in particular, optic atrophy could lead to suspect the syndrome. In our experience optic atrophy was an occasional finding during eyes examination done to recognize the early complications of diabetes. 

Hearing loss can also be an associated pathology of suspicion and it is certainly very interesting the peculiar case that we have described in this paper of a child who was born with congenital sensorineural deafness for which it was necessary to perform the cochlear implant and who he developed diabetes only after a few years which allowed him to make the diagnosis of genetic syndrome confirmed by genetic analysis.

For the same reason, as diagnosis of WS can be confirmed solely on genetic basis, please provide genetic information of your 14 patients in the table.

Response: Thank you for this suggestion. We have added in the table 1 the genetical informations of our patients if present. 

The data in the table right now is largely in Yes/No format, it would however be more interesting if it includes more quantitative characteristics as well. 

Response: we have changed the table 1 and we have included more quantitative characteristics 

In addition, if possible update the data with other features often found in WS patients. Is dysarthria/dysphagia present in your patients? Are they heat intolerant? Do they have problems with the heart? Are there any blood abnormalities beside insulin /HbA1c? 

Please provide a comparison of insulin treatment regime between WS patients and autoimmune T1D patients of same age. How much insulin dose is different? Should this lower insulin dose prompt doctors for genetic testing? 

Response: in the paper we have added news about the insulin regimen and the insulin dose  

Hypoglycaemia is often a problem for insulin recieving WS patients, please include information on how to prevent this. This is especially true for patients recieving GLP1 RA. 

There are more than one drug currently in clinical trials against WS- valproic acid in UK (prof. T. Barett) and dandrolene in US (prof. Urano), perhaps some other drugs as well. Please provide a short note on these drugs as well, are there any good or should they be considered as failed attemts?

Response: Thank you for your suggestions. We have completed the paper with those informations about new drugs suggested by the Reviewer 3 

The abstract is too vague, general and repetitive, i would rather see it rewritten to be more concise. There are some spelling and punctuation mistake that need attention. In general, I reccomend this article for publication after the ammendments

Response: the abstract has been rewritten 

Round 2

Reviewer 1 Report

No comments

Reviewer 2 Report

Thank you for the revision of the article. 
The article has gained in structure and expressiveness as a result. The English is good. I recommend publication